# Self-propagating wave drives morphogenesis of skull bones in vivo

Yiteng Dang [1,2,3], Johanna Lattner [1], Adrian A. Lahola-Chomiak[1], Diana Alves Afonso[1], Elke Ulbricht[4], Anna Taubenberger [4], Steffen Rulands [2,3,5] & Jacqueline M. Tabler [1] ✉

Cellular motion is a key feature of tissue morphogenesis and is often driven by migration. However, migration need not explain cell motion in contexts where there is little free space or no obvious substrate, such as those found during organogenesis of mesenchymal organs including the embryonic skull. Through ex vivo *imaging,* biophysical modeling, and perturbation experiments, we find that mechanical feedback between cell fate and stiffness drives bone expansion and controls bone size in vivo. This mechanical feedback system is sufficient to propagate a wave of differentiation that establishes a collagen gradient which we find sufficient to describe patterns of osteoblast motion. Our work provides a mechanism for coordinated motion that may not rely upon cell migration but on emergent properties of the mesenchymal collective. Identification of such alternative mechanisms of mechanochemical coupling between differentiation and morphogenesis will help in understanding how directed cellular motility arises in complex environments with inhomogeneous material properties.

Tissue morphogenesis requires the coordinated motion of cells to move into regions where they eventually differentiate into various specialized cell types. Such motion is often considered to be driven by cell migration. Cell migration is typically thought to be driven by external gradients breaking tissue isotropy and introducing a preferred direction along which cells migrate. Such gradients can arise from chemical cues (chemotaxis[1]), electrical fields (galvanotaxis[2]), or stiffness (durotaxis[3]). In each example, the motion of cells is driven by the cytoskeletal or membrane dynamics of individual cells, which may be coupled to one another to also drive collective cell migration.

The external stimuli directing cell migration regulate polarized cytoskeletal dynamics that propel cells forward. The ability of cells to move collectively, however, also relies on cell–cell and substrate adhesions. In epithelia, tight cell–cell adhesions allow for the propagation of physical force and cohesive movement[4]. In collective mesenchymal migration, such as that found in the cranial neural crest, where contacts with neighbors repolarize cells toward regions of lower density and substrate interactions provide traction[5,6]. However, as organs are built later in development, mesenchyme will generate contiguous tissues with limited free space between domains that are not always separated by overt boundaries such as a basement membrane (e.g., cranial mesenchyme, which gives rise to dermal, skeletal, and meningeal lineages). Additionally, the migratory potential of mesenchyme must also be restricted if not inhibited entirely to maintain tissue cohesion[7,8]. Therefore, cell migration, as currently described, may be insufficient to explain motility in all contexts involving polarized growth or coordinated dynamics in cell collectives.

In physical contexts, motility can be generated spontaneously through the physical interactions of objects or molecules. Local temperature increases in gases, for example, cause pressure gradients that drive the directed motion of molecules toward cooler regions. Such dynamics can be generated in fluids and within cells as well. At the cellular scale, laser-induced heating of cytoplasm can drive spontaneous cytoplasmic streaming that is sufficient to redirect cellular

[1]Max Planck Institute for Molecular Cell Biology and Genetics, Dresden, Germany. [2]Max Planck Institute for the Physics of Complex Systems, Dresden, Germany. [3]Center for Systems Biology, Dresden, Germany. [4]Biotechnology Center (BIOTEC), Dresden, Germany. [5]Arnold-Sommerfeld-Center for Theoretical Physics, Ludwig-Maximilians-Universität München, München, Germany. ✉e-mail: tabler@mpi-cbg.de

polarity in *C. elegans*[9]. At the tissue scale, spontaneous motion can arise from inhomogeneous proliferation rates, resulting in regions of high proliferation exerting pressure on surrounding cells[10,11]. Similarly, cell motion within intestinal villi is generated by proliferation in the crypt that displaces neighbors due to mechanical coupling through cell–cell adhesions typical of epithelia[12]. In these systems, spontaneous motion is generated from the inhomogeneity of the surrounding physical structure, rather than the intrinsic dynamic behaviors of each object within the system.

To ask whether mesenchymal motility can be generated spontaneously through physical mechanisms other than cell migration, we turn to the skull. The frontal bones or calvaria are formed by bone rudiments that expand medially from the side of the head toward the skull midline[13] between dermal and meningeal layers with which the prospective skull is entirely contiguous. The expansion of these frontal bone rudiments is thought to occur through directed and collective migration of bone-producing osteoblasts toward the top of the head, leaving behind a collagen meshwork that undergoes concomitant mineralization[14-17]. Such a progressive pattern of mineralization generates material inhomogeneities along the axis of expansion. Here, we ask whether such structural inhomogeneity could contribute to bone morphogenesis and whether structure within developing bone could be sufficient to explain the morphogenesis of the skull. Indeed, we find that the feedback between a self-generated stiffness gradient through a wave of differentiation drives cell motion to coordinate the expansion of mesenchymal bones. We discovered a mechanism of mesenchymal cell motion that would not rely on intrinsic cytoskeletal dynamics to drive directed motion through migration, but rather on cell collectives actively shaping and responding to material properties as the tissue develops.

## Results

### Inhomogeneous structure across the axis of frontal bone expansion influences cell motility and division

We first measured the inhomogeneity of tissue material properties and extracellular matrix across the axis of bone growth during peak stages of skull morphogenesis (Fig. 1A; Fig. S1). Atomic Force Microscopy (AFM) and nanoindentation across three locations along the axis of bone extension (Fig. 1B) revealed a stiffness gradient that was highest in the bone center and lowest in the undifferentiated mesenchyme (Fig. 1C; Fig. S2A). Osteoblast differentiation is characterized by enriched expression of fibrillar ECM, such as Collagen 1a1, in bone[18-20]. Second Harmonic Generation (SHG) imaging is a standard label-free method for detecting fibrillar structures such as Col1a1[21-23]. Using SHG, we found that the stiffness gradient we measured with AFM coincides with the enrichment of fibrillar extracellular matrix typical of Col1a1 (Fig. 1C, D). These data demonstrate inhomogeneity in tissue material properties and structure in skull mesenchyme during bone morphogenesis that is consistent with ECM production in differentiated osteoblasts.

To test whether cell motion could be affected by these inhomogeneous tissue material properties, we developed an ex vivo skull imaging system. Entire skull caps were explanted from *Osx1-GFP::Cre* mice, which harbor an osteoblast-specific GFP::Cre recombinase fusion protein[24] (Fig. 1E–G). In this system, cells contiguous with the bone ahead of the front are unlabeled. Although cell membranes are too complex to parse individual cells (Fig. S3A, Supplementary Movie 1), our approach allowed us to distinguish individual nuclei as a proxy for individual osteoblasts, for which motion and cell division could be tracked (Fig. S3B, C; Supplementary Movie 2). We tracked individual nuclei at the osteogenic front separating differentiated osteoblasts from undifferentiated mesenchyme, as well as 200 μm and 400 μm toward the bone center (Fig. 1H). While the mean squared displacement (MSD) of all cells initially scales with time as $t^2$ for all tracked cells, indicating ballistic motion, over time, the MSD of the cells toward the

bone center approaches a linear scaling with time, which reflects diffusive motion (Fig. 1I; S4A). In contrast, the cells in the middle and near the front retain their ballistic motion for much longer, and their velocities remain more correlated over time (Fig. S4B). The diffusive motion of osteoblasts within the differentiated, more densely packed (S2B-D), collagen-rich bone center is consistent with the idea that these cells experience a different stiffness from those at the expanding front. Such MSD patterns suggest that material properties such as stiffness influence features of cell motion, and in particular, restrict a cell's ability to move persistently.

Inhomogeneous tissue-level forces similar to the ones observed here are known to regulate proliferation rates and division orientation in other systems[25,26]. We found reduced proliferation within the bone center, which harbors the most collagen and cells are more densely packed, consistent with inhibited proliferation in densely packed environments (Figs. S2E; S3B, C)[27] as well as terminal differentiation. We also found that labeled osteoblasts divided along the axis of growth in both live imaging and fixed skull caps (Fig. 1J–L; Supplementary Movie 3). Not only were the medial-lateral relationships between daughter cells as well as neighbors maintained for the duration of the movie (Fig. 1M, N; Supplementary Movie 4), but daughter cell displacement was greatest toward the front (Fig. 1O, P). Together, these data support a role for stiffness and collagen gradients in regulating cell dynamics during bone morphogenesis, where the bone front is increasingly compliant when compared to the bone center.

### Inhomogeneous structure is built by progressive osteoblast differentiation ahead of the bone

While tracking individual nuclei, we noticed that cells initially at the osteogenic front, which separates differentiated from undifferentiated cells, were no longer at the leading edge by the end of our live imaging. To confirm this finding, we quantified the relative displacement of the front and compared this displacement to that of individual cells originally residing at the front. We found displacement of the osteogenic front to be consistently greater than that of tracked nuclei (Fig. 2A–C). While we found the expansion rate of the osteogenic front ((15 ± 6) μm) to be comparable to in vivo measurements between E13.5 and E14.5 (Fig. S1D) ((25 ± 4) μm), tracked nuclei moved more slowly compared to the front interface (Fig. 2C). These data suggest that newly differentiated osteoblasts are added to the osteogenic front as the motion of tracked cells cannot explain the extension of the front.

Additional osteoblasts at the osteogenic front could arise from the differentiation of mesenchyme that resides medial to the developing bone. If so, we would expect to find greater GFP intensity in osteoblasts in the bone center compared to those at the front, as GFP signal increases with osteoblast maturation in *Osx1-GFP::Cre* reporter mice. To test this, we quantified GFP intensity in osteoblasts across the mediolateral axis of the bone and indeed found nuclei decreased in signal intensity toward the front (Fig. 2D). Further, this intensity profile was shifted over time, indicating an increase in GFP expression that could arise from newly differentiated cells. To confirm that cells newly expressing *GFP* have only recently activated the *Osx1* promoter and have not lost and reactivated *Osx*-driven expression, we crossed the *Osx1-GFP::Cre* line to *R262RmT/mG*[28] reporter mice. Here, membrane GFP would be driven after accumulation of Cre recombinase in our *Osx1* reporter skulls, and thus, newly differentiated osteoblasts would harbor only nuclear GFP, whereas membrane GFP would be expressed in addition to nuclear GFP in maturing osteoblasts (Fig. 2E–F). Indeed, we found a nuclear-only signal at the osteogenic front where osteoblasts in the bulk of the bone were labeled with both membrane and nuclear GFP (Fig. 2G; Supplemental Video 5). Likewise, we found the expression of an early skeletal marker, *Runx2*, to be expressed ahead of the osteogenic front in *Osx1* negative mesenchyme (Fig. 2H, I). These data demonstrate that differentiation occurs at the osteogenic front and that cell motion is insufficient to explain the dynamics of bone growth.

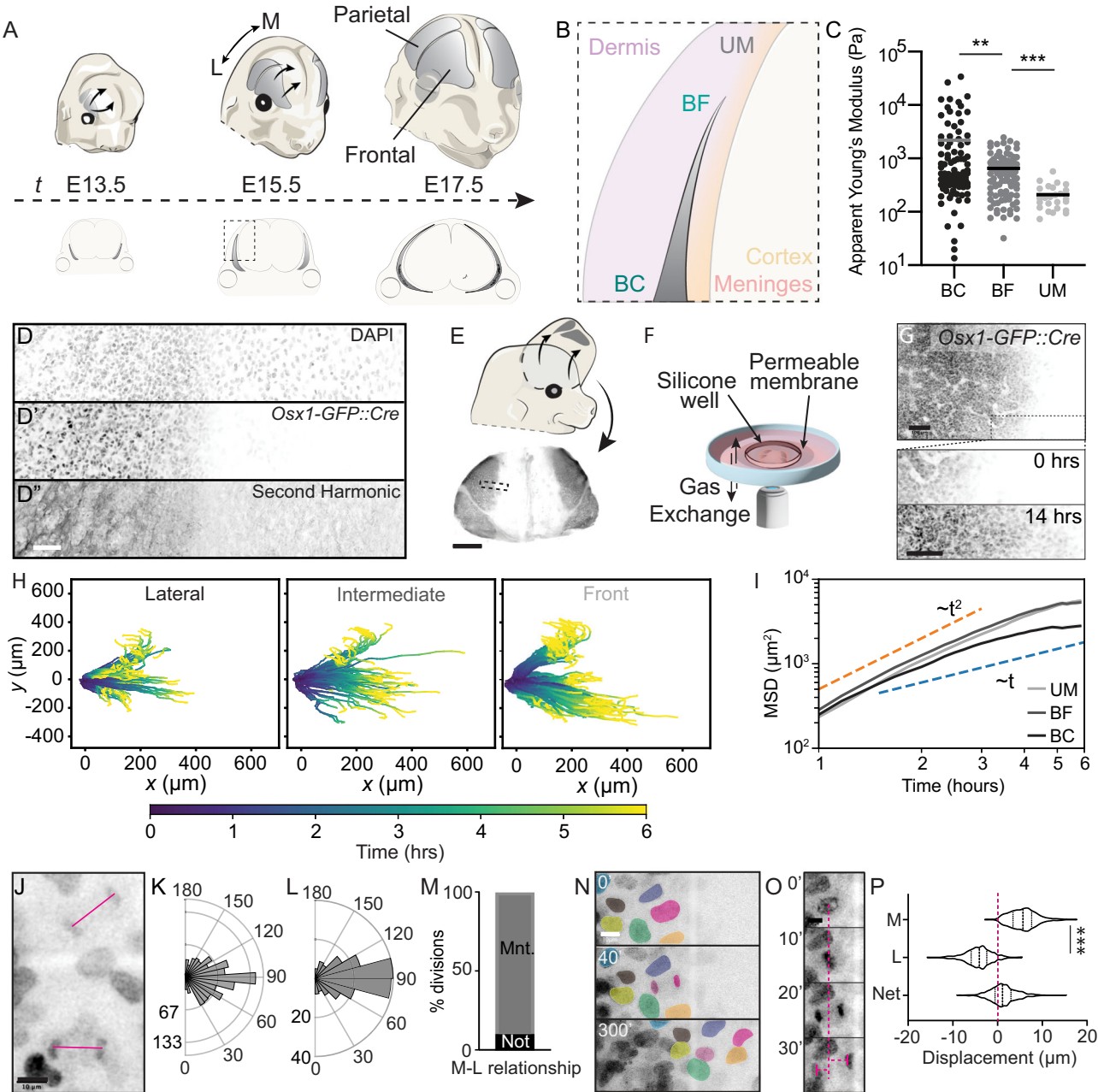

**Fig. 1 | Inhomogeneous material properties and cell behaviors during skull morphogenesis. A** Illustration depicting anisotropic expansion of frontal and parietal bones (gray) toward the top of the head between E13.5, 15.5, and 17.5. **B** Diagram depicting a coronal section of the developing skull with the frontal bones colored in grey and labeled in cyan. Dotted box indicates an inset showing the locations of the bone center (BC), bone front (BF), and undifferentiated mesenchyme (UM). **C** Atomic force microscopy (AFM) measurements of the stiffness (apparent Young's modulus) at E15.5 at the three locations indicated in (**B**). Horizontal bars indicate mean ($N = 8$, bone center $n = 105$, bone front $n = 109$, undifferentiated mesenchyme $n = 25$, Welch's two-tailed $t$-tests, **$p = 0.0017$ for BC vs BF, ***$p < 0.00001$ for BF vs UM. **D** Representative image of nuclear 540 nm autofluorescence generated by 940 nm excitation of the osteogenic front of E14.5 flat-mounted *Osx1:GFP-Cre* skull cap. **D'** Osx1::GFP-Cre fluorescence. **D"** Second Harmonic Generation signal with 940 nm excitation. Scale bar = 50 μm, ($N = 3$) **E** Schematic showing excision of *Osx1-GFP::Cre* labeled skull caps. Scale bar = 100 μm. **F** Diagram depicting ex vivo imaging setup. **G** Max projection of *Osx1-GFP::Cre* labeled frontal bone at 0 h at E13.75. Dotted box indicates insets at 0 h and 14 h shown below. Scale bar = 100 μm. **H** Graph showing example lateral ($n = 139$), intermediate ($n = 117$), and front ($n = 199$) cell tracks from live imaging experiments

combined ($N = 4$), color-coded by time. **I** Graph showing mean squared displacement for osteoblasts in different mediolateral positions, averaged over $N = 4$ images. Shading indicates SEM. **J** Max projection of *Osx1-GFP::Cre* labeled E13.75 live explant showing two late anaphase nuclei with pink lines indicating angle of divisions, which are quantified in (**K, L**). Scale bar = 10 μm. **K** Angle of osteoblast division in E13.75 *Osx1-GFP::Cre* labeled live explants, (Mean angle, 88°, two-tailed Rayleigh z-test = 497.4, $p < 0.00001$, $n = 862$, $N = 8$). **L** Angle of division orientation in fixed *Osx1::GFP-Cre* labeled E13.5 flat-mounted skull caps (Mean angle, 84°, two-tailed Rayleigh $z$-test = 90, 73697178, $p < 0.00001$, $n = 149$, $N = 6$). **M** Graph showing the percentage of osteoblasts that maintain (Maint., gray) or do not maintain (Not maint., black) their mediolateral neighbor relationship ($n = 60$, $N = 4$). **N** Representative stills from *Osx1-GFP::Cre* labeled cultured explants. Osteoblast nuclei have been false colored to demonstrate neighbor relationships reflecting data in (**M**), sister nuclei are overlaid with pink ($N = 4$). Scale bar = 10 μm. **O** Representative time series of an osteoblast division from *Osx1-GFP::Cre* labeled explant showing how nuclear displacement is measured ($N = 8$). Scale bar = 10 μm. **P** Graph showing significant difference in the displacement of medial and lateral sister nuclei after a division (two-tailed two-sample $t$-test, ***$p < 0.0001$, $n = 499$, $N = 8$).

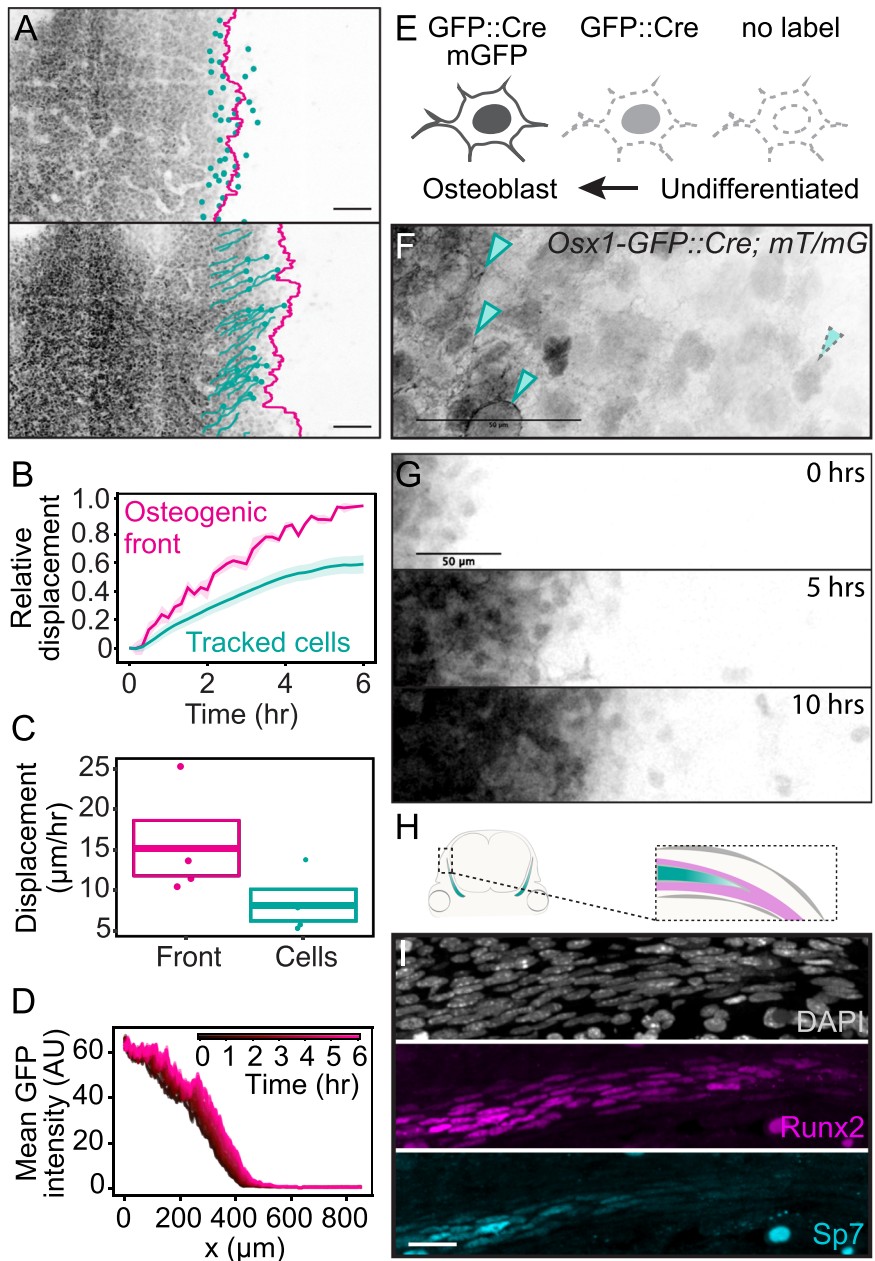

**Fig. 2 | Ex vivo live imaging of bone expansion reveals progressive differentiation of osteoblasts. A** Max projection of *Osx1-GFP::Cre* labeled frontal bone at 0 h (E13.75) and 6 h, together with the osteogenic front (magenta) and individually tracked cells (cyan). Scale bar = 100 μm. **B** The relative displacement for the osteogenic front (magenta) and tracked cells (cyan) over the course of 6 h starting from E13.75, defined as the displacement normalized by the total displacement of the osteogenic front at 6 h. Shaded areas show SEMs (*N* = 4, *n* = 199 tracked cells). **C** Box plot with all individual data points of rate of expansion at the osteogenic front compared to that of tracked cells, computed from the same data as in (**B**). The center represents the mean, and the bounds represent the standard error. **D** Average GFP intensity profiles along the medial-lateral axis for a labeled frontal bone in an ex vivo imaging experiment starting at E13.75. Each graph arises from one time frame, with the color indicated in the legend. **E** GFP labeling scheme in *Osx1-GFP::Cre; R26RmT/mG* explants. **F** Representative fixed tissue image of *Osx1-GFP::Cre; R26RmT/mG* (*N* = 6). The arrows with solid outline indicate cells that have both membrane and nuclear stain, and the arrow with dashed outline indicates a cell with only nuclear stain. **G** Max projections showing GFP localization in E13.75 live movies of *Osx1-GFP::Cre; R26RmT/mG* explants at 5 h time intervals. Scale bars = 50 μm. **H** Schematic of a coronal section with the bone labeled in cyan. The inset shows the imaged areas of I with the approximate domains of Sp7+ osteoblasts (cyan) and Runx2+ precursor cells (magenta). **I** DAPI, Runx2, and SP7 immunoreactivity at the osteogenic front in E14.5 coronal sections (*N* = 8).

## Feedback between tissue material properties and cell fate is sufficient to describe frontal bone morphogenesis

We next asked whether coupling between tissue material properties and cell fate could be sufficient to drive key features of calvarial morphogenesis as observed. To this end, we constructed a minimal theoretical model describing the tissue as a viscous fluid with two types of cells, osteoblasts and undifferentiated mesenchyme (Fig. 3A; Supplementary Text). Our model includes two processes that

modulate local cell concentrations: (1) proliferation and cell death give rise to effective reproduction rates that depend on local cell densities, (2) cell differentiation, whereby progenitors are converted to osteoblasts at a rate that can depend on other (dynamic) variables of the system. Furthermore, we modeled forces capable of generating cellular flows and arising from the balance between pressure gradients, viscous forces, and friction. To model the observed inhomogeneities in stiffness, we let the tissue stiffness depend locally on cell type, to

reflect differential rates of collagen production whereby differentiated osteoblasts generate a stiffer environment than undifferentiated mesenchyme. Conversely, we modeled a differentiation rate that increases with higher stiffness, in accordance with in vitro experiments[29–36], thus generating a mechanical feedback loop between cell fate and stiffness.

Our model generates wave solutions that recapitulate the expansion of the osteoblast domain, with differential velocities for the osteogenic front and for individually tracked cells. To obtain all model parameters, we estimated the stiffnesses and homeostatic cell densities of mesenchyme and osteoblast tissue, as well as the diffusion constant directly from experimental data, and estimated the viscosity, friction coefficient, and relaxation time of the net division rate from literature (Table S1). By fitting a single parameter (describing the relation between differentiation and stiffness) in numerical simulations, we simultaneously fit the experimentally measured values obtained from our live imaging analyses and *Osx1-GFP::Cre* intensity values extracted from fixed labeled stage series skull caps (Fig. 3B; Fig. S5). Moreover, upon fitting only these expansion velocities, the model correctly predicted the spatial profiles of the relative osteoblast concentration (corresponding to GFP intensity) and cell velocity. Specifically, the cell velocity profile shows a peak at the osteogenic front with cells near the front moving faster than cells toward the rear of the bone (Fig. 3C).

Our model also predicted the observation that the spatial profile of osteoblast density is characterized by a stable, expanding wavefront of differentiated osteoblasts (Fig. 3D), as found by quantifying the measured GFP intensity profiles (Fig. 2D). Although such a wavefront may arise in tissues with spatially inhomogeneous cell division rates[37], our PH3 immunostaining revealed no significant differences in proliferation rates between the bone front and undifferentiated mesenchyme (Fig. S2E). Therefore, we proposed that the osteogenic wave is instead driven by the aforementioned mechanical feedback, whereby a self-generated collagen gradient generates both pushing forces from emergent pressure gradients, as well as an osteoblast differentiation wave arising from a stiffness-dependent differentiation rate (Supplementary Text). Further quantification of fluctuations of the osteogenic front confirms that the data are consistent with a biophysical wave with mathematical properties described by our model (Fig. S4C–F; Supplementary Text). Altogether, these data support the idea that a biophysical wave generated by mechanical feedback is sufficient to recapitulate complex tissue dynamics during skull morphogenesis.

### Perturbing the stiffness gradient changes bone size

Our model predicted that a stiffness gradient is sufficient to drive both cell motion and differentiation toward the midline. The velocity of motion and differentiation would then be dependent on the slope of the gradient. The greater the stiffness gradient, the faster cells would move and differentiate. To test this prediction in vivo, we performed two perturbations. In the first, we performed live imaging on E13.75 skull caps in which the stiffer bone center was excised (Fig. 4A). Our model led us to predict that bone expansion should be halted in the absence of the stiff bone center. We found little bone expansion throughout our imaging experiments, suggesting that the stiffer bone center contributes to the morphogenesis of the frontal bone (Fig. 4B). In our second perturbation, we aimed to increase the stiffness gradient, which would increase pushing forces from the bone center to increase the rate of bone expansion. We chemically blocked collagen crosslinking by feeding pregnant dams with Beta-Aminoproprionitrile (BAPN), an irreversible competitive inhibitor of LOX[38–43]. Loss of crosslinking would allow for collagen fiber deformation upon physical interactions with cells or longer-range forces such as stretch from the underlying expanding brain. Indeed, we found that fiber area was perturbed in the newly differentiating bone (Fig. 4F, G), which led to

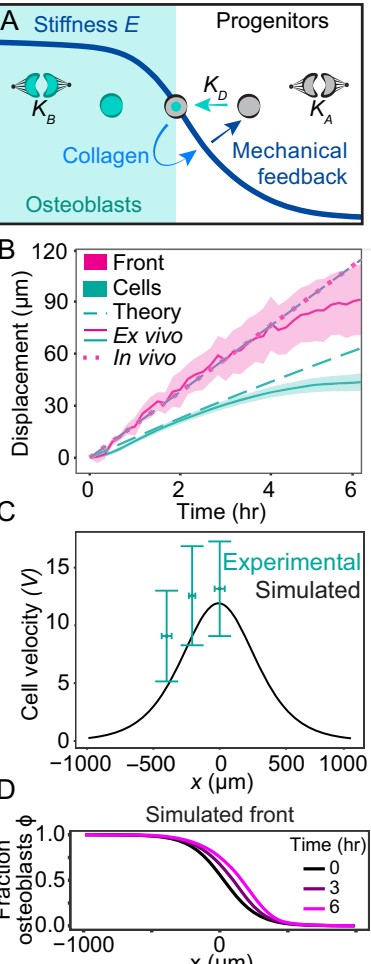

**Fig. 3 | Biophysical model with mechanical feedback between stiffness and cell fate recapitulates imaging results. A** Schematic of the model. **B** Dynamics of the osteogenic front and the tracked cells for the theoretical model, the ex vivo live imaging experiments ($N = 4$), and in vivo fixed tissue images obtained by comparing bone sizes between E13.5 ($n = 7$) and E14.0 ($n = 7$). Shaded areas indicate SEM for the ex vivo data. **C** Cell velocity for the theoretical model compared to the velocities obtained from the ex vivo tracked cells over 6 h (Fig. 1H). Error bars show standard deviations in both horizontal and vertical directions. **D** The simulated fraction of osteoblasts shows a sigmoidal profile across space that travels as a wave. The horizontal axis shows the position along the medial-lateral axis, where the origin represents the initial position of the osteogenic front, here defined to be the location where $\phi = 1/2$.

poorer mineralization at the end of skull morphogenesis, as expected (Fig. S6). As we predicted, BAPN-treatment increased the stiffness gradient by reducing tissue stiffness at the osteogenic front (Fig. 4D–I), although stiffness in the bone center mildly increased (Fig. 4I). Consistent with our prediction, we found significantly larger frontal bones toward the end of skull expansion (Fig. 4J–L). To test whether increased differentiation contributed to these larger bones, we measured the intensity of the Osx1-GFP::Cre reporter as a proxy for osteoblast "age" as before. If more cells differentiate at the front, we would expect cells along the axis of growth to be more similar in differentiation state compared to controls. Indeed, we found that the slope of reporter intensity at the osteogenic front was significantly diminished in BAPN-treated embryos, consistent with increased cell differentiation at the front (Fig. 4M, N). These support our model where a gradient in tissue stiffness or pressure generated by differentiation-dependent collagen production is sufficient to orchestrate frontal bone morphogenesis (Fig. 4O, P).

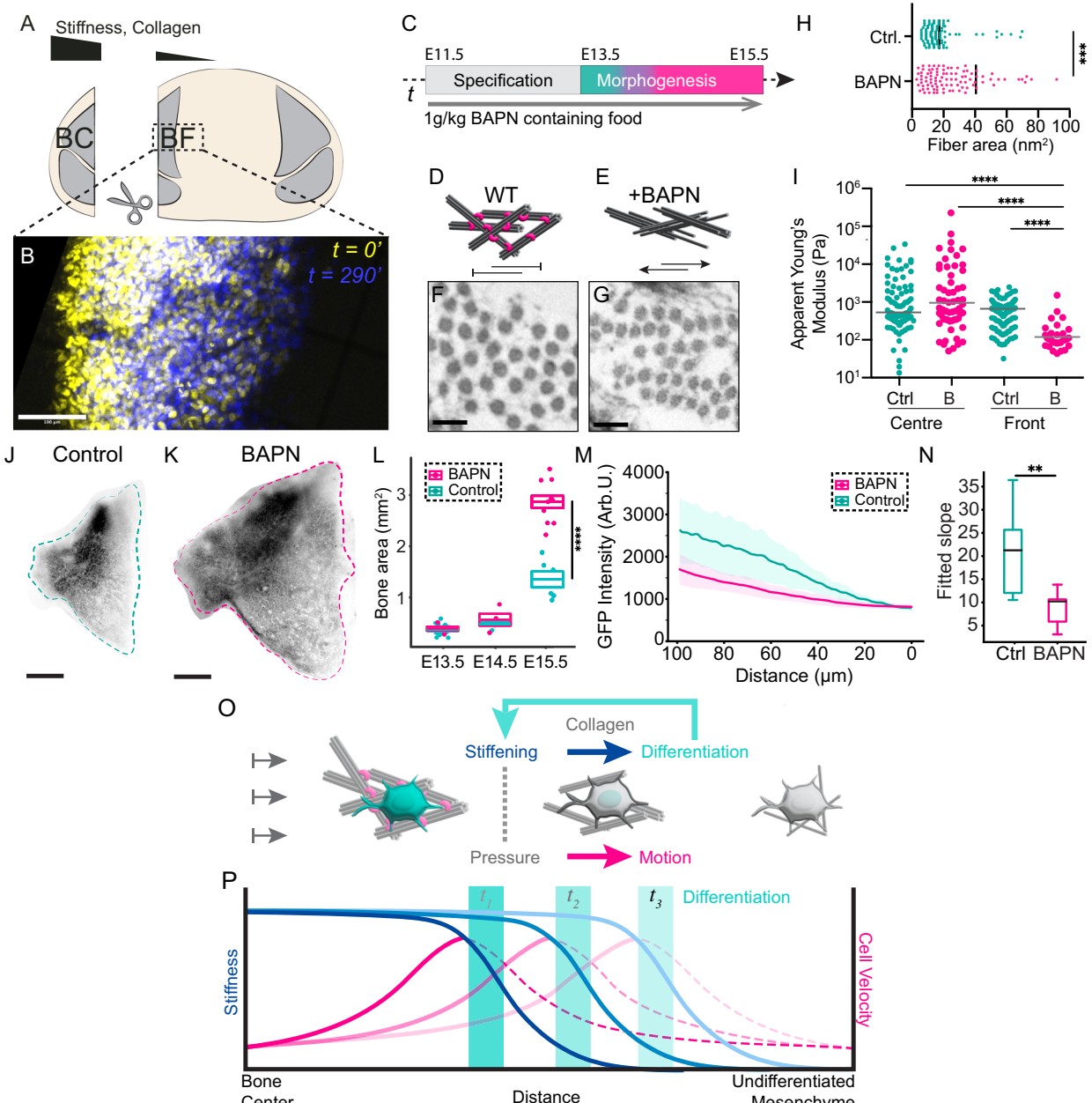

**Fig. 4 | Perturbing collagen crosslinking leads to a larger difference in stiffness across the tissue, resulting in larger bones. A** Schematic showing embryological removal of the bone center from skull explant for live imaging. **B** Stills showing Osx1-GFP::Cre fluorescence of live-imaged skull caps where the bone center has been excised. 0 min (yellow) and 290 min (blue) are overlaid ($N = 5$). Scale bar indicates 100 μm. **C** Schematic of the BAPN feeding protocol. **D** Schematic showing collagen (black) fibrils crosslinked by Lysyl Oxidase (magenta). **E** Schematic showing collagen without crosslinkers in BAPN-treated embryos. **F** TEM image of transected collagen fibrils in differentiating mesenchyme in E15.5 control embryos ($N = 2$, $n = 100$). Scale bars indicate 25 nm. **G** TEM image of transected collagen fibrils in differentiating mesenchyme from E15.5 BAPN-treated embryos ($N = 22$, $n = 103$ (Welch's two-tailed $t$-test, $p < 0.0002$)). Scale bars indicate 25 nm. **H** Violin plot with median (solid line) and quartiles (dashed lines) showing cross-sectional area of collagen fibers in control ($n = 100$) and BAPN-treated embryos ($n = 103$) (Welch's two-tailed $t$-test, $p < 0.0002$). **I** AFM measurements of the stiffness (apparent Young's Modulus) at E15.5 at two different locations of the tissue, for control samples (cyan) and BAPN-treated samples (magenta) ($N = 2$, $n = 409$), with horizontal lines indicating the median (Kruskal–Wallis test, *$p < 0.02$, ****$p < 0.0001$). **J** Representative image of a control *Osx1-GFP::Cre* labeled frontal

bone at E15.5 ($N = 2$, $n = 6$). **K** Representative image of a BAPN-treated *Osx1-GFP::Cre* labeled frontal bone at E15.5 ($N = 3$, $n = 10$). Scale bars indicate 200 μm. **L** Box plot with all individual data points showing measured frontal bone areas for the two conditions at E13.5. The center represents the mean, and the bounds represent the standard error. (WT, $N = 2$, $n = 4$, BAPN, $N = 3$, $n = 13$), E14.5 (WT, $N = 2$ $n = 4$, BAPN, $N = 2$, $n = 4$), and E15.5 (WT, $N = 2$, $n = 6$, BAPN, $N = 3$, $N = 10$) (Mann–Whitney test, $p < 0.0001$). **M** Line plot showing the normalized and aligned GFP intensity profiles at the osteogenic front in fixed control (green) and BAPN-treated (magenta) skull caps at E13.75. The shaded area represents values within one standard deviation of the mean. **N** Box plot showing the fitted slope of GFP intensities between control and BAPN-treated embryos represented in (**M**). The box extends from the first quartile (Q1) to the third quartile (Q3) of the data, with a line at the median. The whiskers extend from the box to the farthest data point lying within 1.5× the inter-quartile range (IQR) from the box. (** represents $p < 0.01$, two-sided Mann–Whitney U-test, $p = 0.00786$). **O** 2D schematic showing a proposed model of anisotropic frontal bone expansion at the level of individual cells. **P** 1D schematic showing proposed model for a self-propagating wave of differentiation, stiffness, and cell motion.

## Discussion

With this work, we suggest that a self-generating wave of differentiation is sufficient to generate cell motion and extend calvarial intramembranous bone. Thus, we propose an alternate model of motility that does not depend upon the intrinsic polarized dynamics of individual cells. Although stiffness gradients are known to drive durotaxis in vivo[44] as well as anti-durotaxis in vitro[45], it has not been shown that such gradients can generate motion without necessitating intrinsic migration. We show how self-organized mechanochemical coupling between stiffness and cell fate could coordinate effective cell motility and further cell differentiation that can, together, ultimately coordinate tissue size. Our work provides a general physical mechanism for generating tissue growth during skull morphogenesis and in mesenchymal tissues more broadly. However, this physical mechanism of collective morphogenesis is challenging to disentangle from the mechanisms that generate polarized migration. Our system does not allow us to observe the membrane dynamics that would help parse migration from displacement suggested here. For example, both cell flows and migration could generate patterns of osteoblast motion seen here, where front cells move more directly compared to rear cells. It is important to note, though, that osteoblasts are not moving on a substrate here rather, they are embedded in the tissue. We also find cellular motion to be down the stiffness and collagen gradient, whereas mesenchymal cells most commonly migrate through durotaxis up stiffness gradients, and the tissue composition is not consistent with fibroblastic anti-durotaxis[3,46]. Further, there are no true leader cells and cell rearrangements are very limited, contrary to what is typical during mesenchymal collective cell migration[1]. Our data can be explained without the need for collective cell migration, which is contrary to previous assumptions[14,15,17]. Instead, our data indicate that differentiation plays a predominant role in establishing the inhomogeneous physical environment that also generates collective motion.

At the molecular level, distinguishing between active migration and our differentiation-mediated motility is prohibitive. The same cytoskeletal regulators that control migration are also required for transducing mechanical information that mediates differentiation as well as proliferation[47–49]. Therefore, we chose to perform physical perturbations that altered stiffness gradients without interfering with intracellular mechanotransduction. Although we find here and in other recent work that collagen organization controls cranial mesenchymal fate (preprint,[50]), how the molecular mechanisms that mediate mechanosensation, differentiation, and collagen production regulate the self-propagating morphogenesis we find here remains open.

Previous in vitro studies have established that both osteoblast differentiation and morphogen signaling increase with substrate stiffness[51–53]. However, here we find the stiffness values measured in our study are an order of magnitude lower than those typically required for osteoblast differentiation in vitro. These data indicate that mechanical regulation of cell fate in vivo or within cell collectives remains unclear. Thus, our work also highlights the need for in vivo studies to confirm mechanisms found in vitro and to establish physiological parameters in developing tissues, which cannot always be directly extrapolated from in vitro studies.

Additionally, our perturbation experiments also suggest that the slope of the stiffness gradient determines bone size, in accordance with our biophysical theory. Understanding how differences in spatial gradients of tissue stiffness influence differentiation likelihoods, kinetics, or morphogen signaling also remains to be explored in vivo. However, if the slope of the stiffness gradient is important in regulating differentiation, such a feature could function as a tunable parameter controlling overall skull size and enable the plethora of skull sizes in different organisms. Further, our work demonstrates that conceptual mechanisms driving molecular motion in non-living physical systems also extend to cell motility at the scale of tissues.

Together, this work extends physical principles that regulate dynamics at the molecular and subcellular scales to those of tissues. We provide a conceptual framework to understand morphogenesis and motion in biology that emerges within collectives. In asking whether cellular mechanisms such as durotaxis found in vitro are needed to describe morphogenesis in the densely packed cells found in tissues, we must also consider other physical processes or mechanisms. Understanding alternative modes of motion, as suggested here, has wider implications for engineering tissues in vitro and interrogating mechanisms of morphogenesis in vivo.

## Methods

### Ethics and inclusion

This work was performed in accordance with The TRUST Code for A Global Code of Conduct for Equitable Research Partnerships. This work has involved local and foreign researchers throughout the project as part of a collaboration with local partners. Research responsibilities were agreed upon within the group, and experiments were performed in accordance with local government guidelines for safety and animal experimentation. This work was conducted within the local research environment and was well-supported by the ethics and safety infrastructures at the MPI-CBG and Dresden CONCEPT institutes.

Experiments conducted at the University of Texas, Austin, were performed in accordance with approved IACUC protocols. Experiments conducted at the MPI-CBG adhered to the German Welfare Act and were overseen by the Institutional Animal Welfare Officer under licenses TVV22/2020 and TVV42/2021.

### Mouse lines

The following mouse lines were used: *Sp7-mCherry* (Tg(Sp7/mCherry) 2Pmay/J (Su, available via Jax))[54], *Osx1-GFP::Cre* (B6.Cg-Tg(Sp7-tTA,-tetO-EGFP/cre)1AMC/J, available via JAX)[24], *R26RmT/mG* (GT(Rosa) 26Sortm4(ACTB-tdTomato-EGFP)Luo, available via JAX)[28] and C57Bl/6JOlaHsd or C57Bl/6NTac. Genotyping was performed as described in the original publications. Lines used are reported within the results according to the experiment. Mice were held in individually ventilated plastic cages with free access to food and water. The dedicated cage-level ventilation ensures an optimal environment with stable temperature and humidity, as well as a constant supply of clean fresh air. Each cage has a thick layer of aspen bedding, as well as a combination of different materials (wooden wool, cotton nestlets) that mice like to use to build nests. Mice were housed in small family groups.

### Lysyl-oxidase inhibition

For collagen crosslinking inhibition studies, pregnant *Osx1-GFP::Cre* females were fed 0.25% (1 g/kg) BAPN-containing food from E11.5, the onset of skull morphogenesis. Control animals were instead fed with vehicle-loaded control food. For embryo collection, pregnant females were euthanized by cervical dislocation, and embryos were collected for downstream analysis. The protocol was approved by the Institutional Animal Welfare Officer and the local. BAPN-containing food made by Safe Complete Care Competence.

### Culture media

High Glucose DMEM (Sigma D6469) is supplemented with 10% Fetal Calf Serum (Gibco A3160502), 100 mg/mL ascorbic acid (Sigma, PHR1008), 10 μM beta-glycerolphosphate (Sigma 50020), and 0.5 mL in 50 mL 100× Antibiotic Antimycotic (Sigma 15240062).

### Imaging chamber preparation

Sarstedt Lumox dish 35 chambers were coated in Fibronectin from bovine plasma (Sigma F1141-1MG) diluted 1:1 with DMEM. (Sigma D6469). Dishes were left to dry (~15 min) while embryos and media were prepared. A cyclopore polycarbonate membrane filter (GE Healthcare Whatman, 7060-2516) and silicone well (Flexiperm ConA,

**Table 1 | Antibodies and dilutions**

| Antibody | Manufacturer | Catalog Number | Dilution |
|---|---|---|---|
| Rabbit anti-Osx1 | Abcam | ab22552 | |
| Mouse anti-Runx2 | Santa Cruz Biotechnology | sc-390715 | 1:50 |
| Mouse monoclonal a-PH3 | EMD Milipore Corporation | MERK 06-570 | 1:400 |
| Alexa Fluor 488 Goat anti-Mouse IgG | Invitrogen | A-11001 | 1:500 |
| Alexa Fluor 568 Goat anti-Rabbit IgG | Invitrogen | A-11036 | 1:500 |

Sarstedt, 96077434) for holding the sample in place were placed in DMEM after being washed in 70% Ethanol in advance of sample preparation.

### Immunohistochemistry

All immunohistochemistry stainings were performed according to standard protocols. All embryos were collected in cold PBS and fixed in 4% PFA. After fixation, embryos were embedded in 15% sucrose/7.5% gelatin and frozen in dry ice. 35 μm coronal sections were collected for nuclear shape and nuclear envelope analysis. For immunohistochemistry antibody staining, 20 μm coronal sections were collected. Antigen retrieval was performed in sodium citrate buffer (pH 6.0). Sections were blocked with 10% goat serum in PBS for 1 h at room temperature and incubated with primary antibodies overnight at room temperature. All washing steps were performed in 1× PBS. A list of the used primary antibodies can be found in Table 1. Coronal sections were then incubated with secondary antibodies for 2 h at room temperature, counterstained with DAPI, and then coverslipped with Vectashield (Vector Labs, H1000).

### Regional proliferation from sections

20 μm coronal cryosections were obtained from WT embryos and stained for Ph3, Sp7, and DAPI. Ph3-positive cells were counted in the undifferentiated mesenchyme 100 μm ahead of the last Sp7-labeled cell at the osteogenic front, 100 μm behind the front, and a 100 μm box was selected from the thickest region of the Sp7-labeled bone. The Cell counter jar plugin in Fiji was then used to count DAPI-labeled nuclei and Ph3-positive nuclei in these regions, and the percentage of Ph3-positive cells was then calculated.

### Embryo manipulation for osteoblast live imaging

E13.75 embryos are extracted from yolk sacs and scored for reporter gene expression in DMEM (Sigma D6469). Skull caps comprising epidermis, the paired frontal and parietal bones, and meninges were excised with Vannas spring scissors 2.5 mm (F.S.T,15001-08), as indicated in Fig. 1B, in fresh DMEM. The skull explant was transferred to the center of the preprepared gas-permeable dish, meningeal (basal) side down, with the convex side of a 130 mm double spatula. The DMEM-washed membrane is dried slightly by dabbing it once on tissue paper. The membrane was then gently laid down over the sample, being careful to avoid bubbles. Next, the washed silicone well was dabbed dry with tissue paper and quickly placed on top of the permeable membrane such that the sample sits underneath the center of the well. The dish was then filled with 2.5 mL culture media, followed by a thin layer of mineral oil (Sigma M-8410). The sample was then transferred to the environmental chamber of the microscope.

### Flat-mount skull cap imaging of fixed samples and quantification

After excising and fixing, skull caps were mounted flat in vectashield and imaged using the Zeiss AxioZoom ApoTome system. The obtained bone fronts were quantified by first obtaining intensity profiles along the axis of expansion using the ImageJ/Fiji function "Plot Profile". To align the obtained profiles with respect to the osteogenic front, we first determined the baseline intensity by averaging the intensity across a region of the undifferentiated mesenchyme. Then we define our alignment point to be the position where the intensity first exceeds 10% above the baseline value, and measure distances relative to this point. We quantify that increases in intensity are obtained by fitting a linear function using the least squares method to the intensity profiles in a region between 0 and 100 microns to the alignment point and comparing fitted slopes. To obtain average intensity values across samples in this region, for each sample, we binned the distances into bins spaced 1 micron apart for each of the samples, averaged over the values within each bin for a given sample, and then averaged these values from different samples.

### Alizarin red staining

After fixing Embryos in 4% PFA overnight at room temperature, embryos were placed in 1× PBS, and extraembryonic membranes were removed. Embryos were then fixed in 95% ethanol for 1 h. Embryos were then placed in acetone overnight at room temperature. Acetone was then replaced with Alizarin red staining solution (0.005% in 1% KOH - 10 mg Alizarin red in 200 ml 11% KOH), and embryos were incubated on a rotating shaker for 3 h. After, embryos were washed overnight in 1% KOH and then transferred to 50% glycerol: 50% 1% KOH solution and incubated at room temperature until tissue appeared transparent. Once cleared, embryos were transferred to 50% glycerol: 50% ethanol.

### Flat-mount skull cap imaging of stained samples

After attaining, skull caps were mounted flat in 50% glycerol: 50% ethanol and imaged using the Zeiss AxioZoom ApoTome system.

### Second harmonic generation

*Osx1::GFP-Cre* skull caps were excised and mounted as before and imaged using a 2-photon excitation system of the Leica DMI 4000 and Olympus UplanSApo 40×/0.90 Dry objective. 488 laser lines were used to excite the GFP::Cre fusion protein, and 950 nm for second harmonic generation. We also collected simultaneous images of Nuclear autofluorescence at 490 nm using 950 nm excitation.

### Live imaging

Our explant system is adaptable to almost any inverted confocal microscope with an air objective and resonant scanner. Three different confocal microscopes were used in this study. Movies were captured on the single-photon Nikon TiE system with a CFI60 Plan Apochromat Lambda 40× Objective Lens. Next, we used a 2-photon excitation system of the Leica DMI 4000 and Olympus UplanSApo 40×/0.90 Dry objective. 488 laser lines were used to excite the GFP::Cre fusion protein. Later, explants were imaged Andor Revolution WD Borealis Mosaic (Andor) Spinning Disk confocal using the Olympus UplanSApo 40×/0.90 Dry objective.

### Live imaging stitching

To facilitate downstream analysis of multiple position acquisition time-lapse imaging data, we used an ImageJ/Fiji[55,56] script developed by the Max Planck Institute for Cell Biology and Genetics Scientific Computing Facility (https://github.com/PreibischLab/BigStitcher/). The script converts each tile of the raw multiphoton imaging data set into the

TIFF image format, then uses the BigStitcher plugin for ImageJ/Fiji to stitch the tiles[57].

## Manual live imaging analysis

To facilitate image analysis using tools available in the base ImageJ/Fiji package, a maximum intensity projection of live imaging was generated using the Z-projection plugin. To obtain *Osx1-GFP::Cre* intensity measurements along the medial-lateral axis, we projected and averaged the intensity of these maximum intensity projected images over the vertical axis.

## Transmitted electron microscopy

WT and BAPN embryos were collected as described above and fixed with 5% glutarladehyde/1% tannic acid in 0.1 M PBS pH 7.2. Fixed heads were embedded in 4% low-melting agarose, and 200 μm sections were obtained using a vibratome (Leica, VT1200S). Sections were post-fixed with 1% osmium tetroxide (Electron Microscopy Sciences; Cat# 19190) in water. Sections were dehydrated in serial steps (30%, 50%, 70%, 80%, 90%, and 100%) of Ethanol (EtOH), infiltrated with 1:3 EPON LX112/EtOH, 1:1 EPON LX112/EtOH, 3:1 LX112/EtOH, and pure LX112. Sections were embedded on Teflon-coated slides with Aclar spacers (7.8 mil, Science Services and Miller-Stephenson; ordering numbers: DF-3R, respectively). Ultra-thin cross-sections (70 nm) were obtained using an ultramicrotome (Leica, FC7). Sections were post-stained with uranyl acetate (Electron Microscopy Sciences; Cat# 22400) and lead citrate (Electron Microscopy Sciences; Cat# 17800) and viewed in a Morgagni (FEI/Thermoscientific) transmission electron microscope, equipped with a Morada CCD camera (Emsis), at 80 kV.

## Fiber quantification

For collagen fibril size estimation, only fibrils aligned in the AP axis were considered. Firstly, the outline of at least 380 fibrils was manually drawn using the *Freehand selection* tool in Fiji[55]. Cross-sectional fiber areas were calculated using the *Measure* plugin in Fiji.

## Atomic force microscopy

For measurements of tissue bulk stiffness embryos were collected in cold 1× PBS, heads were dissected and embedded in 4% low gelling agarose (Sigma, A4018). 2 mm sections were obtained using a vibratome (Leica, VT1200S) and immobilized using tissue seal (histoacryl blue) in a polystyrene-bottom dish (TPP, 93060). Measurements were performed using a Cellhesion 200 (JPK Instruments/Bruker) mounted on top of a Zeiss Axio Zoom (Zeiss, V16). The cantilevers (arrow T1, NanoWorld), modified with 20 μm diameter polystyrene beads (microparticles GmbH), were calibrated by the thermal noise method using built-in procedures of the SPM software (JPK Instruments). Measurements were performed at room temperature (18−20 °C). Individual force–distance curves were acquired with defined approach and retract velocity (7.5 μm/s) and with contact forces ranging from 2.5 to 10 nN in order to reach approximately constant indentation depths of 2 μm. At least five specimens were probed for each tissue in a 25 μm grid at 5 μm intervals. The apparent Young's modulus $E$ was extracted from approach force–distance curves by fitting to the Hertz/Sneddon[57,58] model for a spherical indenter using JPK data processing software. Measurements were tested for normality using D'Agostino and Pearson's test using Prism. As we found the distributions to have different skewness, we transformed the data and performed a Welch's *t*-test to determine significant differences between each condition.

## NanoIndentation

Nanoindentation was performed using the Chiaro Nanoindentor system (S-Chairo-ST, third generation). Samples were plated on a gas-permeable membrane, and for measurements, media was removed. Samples were viewed using Zeiss Inverted CCD and 5×/0.15 Plan-Neofluar, Air, Ph1, Zeiss objective. Probes with a tip radius of 22 μm and stiffness of 0.41 N/m were used to measure bone, while probes with a tip radius of 56 μm and 0.033 N/m stiffness were used to measure the bone front and undifferentiated mesenchyme. Contact points were defined automatically, and load-indentation curves were fitted using the Hertzian model.

## Cell tracing and quantification of cell tracks

The Manual Tracking plugin in ImageJ/Fiji was used to record the motion of osteoblasts at different positions in the bone. Tracks were recorded for osteoblasts at three different positions across the bone: (1) osteoblasts at the osteogenic front, (2) osteoblasts approximately 200 μm lateral to the front, and (3) osteoblasts approximately 400 μm lateral to the front.

From the cell tracing, we obtained individual cell tracks consisting of sets of two-dimensional positions $(\vec{x}_i(t), \vec{x}_i(t))$ for $1 \leq i \leq N$ at different times $0 \leq t \leq T$. From this, we directly computed the average cell velocities $\overline{V}_i = \frac{1}{T}|(\vec{x}_i(T) - \vec{x}_i(0))|$ and mean squared displacements $\frac{1}{N}\mathsf{L}_{i=1}^{N}|(\vec{x}_i(T) - \vec{x}_i(0))|^2$. To compare directions of motion, we computed a spatial correlation function between normalized velocities of cells, defined as

$$C_{VV}(|y_i - y_j|, \text{t}) = \langle v_i(\text{t}), v_j(\text{t}) \rangle.$$

Here, $y_i$, $y_j$ indicate the y-positions of cells $i$ and $j$, $v_i(t)$, $v_j(t)$ their normalized velocities at time $t$, and the brackets indicate averages over all cells at a distance $|y_i - y_j|$ from each other (in practice obtained from binning cells with comparable distances to each other).

## Division orientation

Divisions were identified by observing metaphase plate formation and anaphase on subsequent time frames. Using the centroids of each daughter cell, the angle of divisions was determined using the ImageJ/Fiji ImageJ/Fiji angle tool relative to the angle of the osteogenic front. In this measurement scheme, 90° represents a cell dividing parallel to the mediolateral axis of expansion. This degree was then converted to radians. The mean division angle, theta, was calculated by deriving r using circular statistics (ACAN) functions in Excel.

## Daughter displacement

The distance that each daughter cell displaced along the axis of expansion was measured using the ImageJ/Fiji line tool. The centroid of each daughter cell was measured relative to the position of the metaphase plate centroid along the axis of expansion. To test for significant differences in distal versus mediolateral displacement, we first tested whether our data were normally distributed using D'Agostino & Pearson's tests using Prism. We found our data to be nonuniform in each condition and therefore used a Wilcoxon matched-pairs signed-rank two-tailed test.

## BAPN protocol

1 g/kg (Sigma Aldrich, A3134) containing food was fed to pregnant dams from E11.5 until collection at E15.5.

## Quantification of bone sizes

We obtained fixed tissue images of the *Osx1::GFP-Cre* and *Sp7-mCherry* lines for WT and BAPN at equal magnifications for E13.5, E14.5, and E15.5. We first segmented the images to extract the region corresponding to the frontal bone in each image. We did this by thresholding the images using a common intensity threshold, and then selected the largest connected region corresponding to the frontal bone for each image.

## Reporting summary

Further information on research design is available in the Nature Portfolio Reporting Summary linked to this article.

## Data availability

The authors confirm that the data supporting the findings of this study are available within the article and its supplementary materials. Source data are provided with this paper.

## Code availability

Code developed for this work and modeling therein are publicly available through the GitHub repository https://github.com/YitengDang/SkullWave and are also included as part of the Zenodo repository https://zenodo.org/records/12960584. Within the repository, a readme.txt file explaining the main structure of the repository can be found. The Code subfolder is a copy of the GitHub repository at https://github.com/YitengDang/SkullWave and contains all code developed and used during this project. The Data subfolder contains data used to generate the main plots of the paper. The README.md file of the GitHub repository contains a further description of the various subfolders and files found in the repository. Finally, detailed information on running the simulations has been added to the Mathematica script at./Code/For publication/Tissue model simulator.nb.

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

## Acknowledgements
We thank the Biomedical Services Facility, the Light Microscopy Facility, and the Electron Microscopy Facility at the MPI-CBG for assistance with data collection and optimization of imaging protocols. We thank Jakob Pyszkowski and Optics11 for their support with nanonindentation measurements. We thank members of the Tabler Lab, Stephan Grill, Wieland Huttner, Michele Marass, Anäis Bailles, Ricard Alert, Marko Brankatschk, Arthur Boutillon, and Jan Brugués for helpful discussions and critical feedback. This work was funded in part by the DFG (TA1515/1- 1), the National Institute of Dental and Craniofacial Research/NIH (F32DE023272), and core funding from the MPI-CBG. Y.D. was funded in part by an ELBE Fellowship (MPI-CBG and CSBD). We are deeply thankful to Karen Liu and John Wallingford for inspiring this work and for their instrumental support in its development.

## Author contributions
Conceptualization: J.M.T. Methodology: J.M.T., Y.D. A.T., S.R., and E.U. Investigation: J.M.T., Y.D., A.L.C., D.A.A., J.L., A.T., and E.U. Visualization: J.M.T. and Y.D. Supervision: J.M.T. and S.R. Writing: J.M.T. and Y.D.

## Funding

## Competing interests
The authors declare no competing interests.
