## [Transparent Peer Review file · Nature Communications]

Self-propagating wave drives morphogenesis of skull bones in vivo

Corresponding Author: Dr Jacqueline Tabler

Version 0:

Reviewer comments:

Reviewer #1

(Remarks to the Author)
NCOMMS-24-14942

The revised manuscript titled "Self-propagating wave drives noncanonical antidurotaxis of skull bones in vivo" has been significantly enhanced with additional data supporting the findings and conclusions. The authors have addressed most of the comments with either sufficient data or detailed explanations. The manuscript has generally improved, and I have a few minor suggestions:

1. The authors responded well to Comment 1 with a clear explanation and additional single-cell data. It would benefit readers who are not familiar with Second Harmonic Generation (SHG) if this information were included in the manuscript. Additionally, incorporating the single-cell data within the manuscript would greatly enhance its clarity and comprehensiveness.
2. The demonstration of the interaction between stiffness and classic signaling pathways (regarding Comment 4) is intriguing. Incorporating downstream signaling and crosstalk would substantially strengthen the mechanistic understanding presented in the manuscript. While I understand the authors' intention to reserve some data for more in-depth analysis in future work, it is crucial to include some mechanistic analysis in the current manuscript to enhance its impact and completeness.

(Remarks on code availability)

Reviewer #3

(Remarks to the Author)

The editor has invited us to comment specifically on the mathematical modeling in the manuscript, welcoming comments on other aspects of the manuscript. Overall, the authors have derived, analyzed, and implemented a useful and appropriate model for analyzing self-propagating wave of bone morphogenesis. Firstly, they show how the equations can be derived from cell dynamics and mechanics. Then they use knowledge on the FKPP wave to derive conditions on the parameters that would produce a wave. The parameters are obtained from the experimental data and literature.

Detailed comments for sections:

Theoretical model: The derivation looks all right, but can be written more clearly. Please be clear when referring other parts of the text, (e.g. "as described below" on line 630), and we found some typos that need correction (like velocities v_A and v_B).

The mixing of variable names A and B on the one hand and mnemonics derived from the cell names "osteoblasts" and "mesenchymal cells" (e.g., E_M and E_O) made the model difficult to read at some places. It would be helpful to use similar

mnemonic symbols for A and B, e.g. , c_O and c_M.

Please explain the rationale for using Fick's law of diffusion with respect to the cells B.

We felt that most attention should go to the section 'Wave Solutions'.

- Could you explain why "osteoblast expansion dynamics follows that of a linear unstable wave", l. 679? Even if you came to this conclusion based on observations in the experiments, you cannot a rigid mathematical claim like this. An unstable wave is, as the name says, not stable and therefore not observable. The source that is cited, (67), discussed wave propagation towards an unstable homogeneous steady state, which does not imply that the wave itself is unstable. These comments also refer to the first few sentences in section "Wave Solutions": Not every system that shows a wave is FKPP.

- Two mechanisms are described that are "capable of generating a FKPP wave". This wording indicates that only having "(1) a difference in proliferation rates" would be enough to create a FKPP wave, however it is necessary that $f(\phi=0)=0$, so k_D needs to be dependent on ϕ . Consider starting with the conditions that are necessary for a FKPP wave (l. 688-689) and follow up with a mathematically rigorous approach that derives possible mechanisms that can generate such a wave. We think it would be wise to approach it the following way:

FKPP wave iff $f(1)=f(0)=0$, $f'(0)>0$ and $f'(1)<0$

$f(1)=0$ is already correct

So, we need

$f(0)=k_D(E(\phi=0))=0$

$f(\phi)=(k_B-k_A)(1-\phi)-(k_B-k_A)\phi+(1-\phi)k_D-k_D$

so

$f(0)=(k_B-k_A)+k_D(0)$

$f(1)=-(k_B-k_A)-k_D(1)$

- Lastly, it may be better to move section "Wave Solutions" to after section ("Full System"), because you need the derived equations and information like boundary conditions to do your FKPP analysis.

Comments on other aspects of the manuscript:

- At the end of the first introductory paragraph, the authors write: "Indeed, the suffix -taxis which means directional movement, is entirely synonymous with cellular migration in cell and developmental biology." We do not agree with this statement. Taxis specifically means directional migration along environmental (or self-generated) gradient, as the authors state in the previous sentences. However, cellular migration also includes *random* cell migration (e.g. of neural crest cells or immune cells) and directed cell migration *not* along a specific gradient (sometimes also seen in collective neural crest cell migration, or e.g. in crescent shaped keratocytes). This remark also applies to l. 39.

- Please integrate the mathematical model better within the main text. The part on modeling starts with (l. 149) "We simulated the model with realistic biological parameters", followed by the model predictions. Please clarify on what biological assumptions the model is based, and how these related to the experimental findings. We recommend that the process descriptions (l. 629-633) as well as the mechanical feedback mechanism (l. 644 and onwards) given in the supplementary material are moved to the main text. Then later on in the main text the paragraph starting with 'Therefore, we proposed that the osteogenic wave of

our model (Fig. S4C-F; Supplementary Text) ...' could be made more specific by highlighting specific predictions on the role of mechanical feedback and discussing how exactly these agree (or not) with the data.

- In section "Perturbing the stiffness gradient changes bone size" we would prefer that the model predictions are shown first (after l. 175) after which it can be appreciated to what extent the model predictions match the data. Currently, Fig. 4O-P do not give sufficient specific information as to whether and how the model predictions match the data (i.e., where can we see in this figure that 'Our model predicts that a greater difference in tissue stiffness between the bone center and front would promote the expansion of frontal bones.'?). Possibly this could be solved by a more detailed description in the caption and reference to figure 4P after the statement 'our model predicts that...'

(Remarks on code availability)

Reviewer #4

(Remarks to the Author)

(Remarks on code availability)

Version 1:

Reviewer comments:

Reviewer #3

(Remarks to the Author)

The authors have significantly clarified the integration of the mathematical model in the main text, and they have improved the supplementary text. In this new version, the story has become stronger from a mathematical viewpoint. There are still a few small corrections to be made:

ll. 838-840:

Add: "Since $E_B > E_A$ "

"Above constraints": please clarify to what constraints this refers to.

Add some information when you introduce new variables or parameters. What is α and what conditions does it have to comply with? (e.g. $\alpha > 0$)

(825- 829) add "," and "." after equations

(864) Change " E_O " and " E_M " to the new notation E_A and E_B and check the rest of the text for remaining instances.

(Remarks on code availability)

There is a data availability statement, but it is disturbingly inadequate. The authors need to have a look at it. They write that the code is available "through the Zenoba (sic) code repository". (Googling 'Zenoba' bring me to a brand of sunscreen). The link brings us to the bioRxiv-preprint of this manuscript, where we did not find any code.

After some googling, we found a Zenodo repository at <https://zenodo.org/records/8246065>. This repository contains data and Mathematica code, but no instructions. We tried to run two of the Mathematica-files without success.

'Tissue_expansion_model_4_FD_solver_loop.nb' starts by attempting to run some data from one of the authors home folders. We briefly looked into changing the 'LoadFolder', but it is quite unclear to what folder we need to point LoadFolder, because the in the Zenodo 'Data' folder do not align with those mentioned in the Mathematica File. There is no 'simulation_set1/finished_sims_A/data' and in the existing folders we find files mathcing '*_phi_*', but none matching '*_V_*' or '*_rho_*'.

We also checked 'Tissue_expansion_simplified_model_2022_12.nb'. The code was hard to read or to align with the equations in the manuscript. We attempted evaluating all expressions up to 'ndsol = NDSolve' (which is expected to numerically solve the actual model), but this expression failed with "NDSolve::ndcf: Repeated convergence test failure at t == 0.0004; unable to continue."

We would kindly ask the authors to provide a 'README' file, including some instructions for running the model and reproducing the modeling results presented in the manuscript (i.e. indicating specifically for the modeling figures how they can be reproduced).

Reviewer #4

(Remarks to the Author)

(Remarks on code availability)

Version 2:

Reviewer comments:

Reviewer #3

(Remarks to the Author)

The authors have satisfactorily address all our comments. The code works fine too now, thank you!

(Remarks on code availability)

Reviewer #4

(Remarks to the Author)

(Remarks on code availability)

RESPONSE TO REVIEWER COMMENTS NCOMMS-24-14942-T

Reviewer #1 (Remarks to the Author):

NCOMMS-24-14942

The revised manuscript titled "Self-propagating wave drives noncanonical antidurotaxis of skull bones in vivo" has been significantly enhanced with additional data supporting the findings and conclusions. The authors have addressed most of the comments with either sufficient data or detailed explanations. The manuscript has generally improved, and I have a few minor suggestions:

We thank the reviewer for their help in improving our manuscript.

1. The authors responded well to Comment 1 with a clear explanation and additional single-cell data. It would benefit readers who are not familiar with Second Harmonic Generation (SHG) if this information were included in the manuscript. Additionally, incorporating the single-cell data within the manuscript would greatly enhance its clarity and comprehensiveness.

We are pleased to have clarified our use of SHG and have further added to the text to justify its use here (see text below).

We have chosen not to include the single cell data shared with the reviewers previously. The significant over representation of *Col1a1* expression in differentiating osteoblasts is well established in the field, making the added insight from these data somewhat redundant. As these data are complex in terms of cell types present and the necessary statistical analyses, adding these data would require a complete overhaul of the manuscript. We feel that the primary physical mechanism which we wish to communicate in this work would be lost amongst several figures explaining our single cell data. By adding these data which represent a five year effort, we would also be disadvantaging my trainee who is preparing this work for publication where the biological insights are significantly greater than would be added here. We hope that the additional references clarify the use of SHG in imaging collagen generated by osteoblasts sufficiently to support our claims. We have changed a sentence to make the enriched expression of collagen in osteoblasts more overt:

Lines 74-79: Osteoblast differentiation is characterized by enriched expression of fibrillar ECM such as Collagen 1a1 (18-20). Second Harmonic Generation (SHG) imaging is a standard label-free method for detecting fibrillar structures such as Col1a1 in bone (21-23). Using SHG, we found that the stiffness gradient we measured with AFM, coincides with the enrichment of extracellular matrix typical of Col1a1 (Figure 1D).

2. The demonstration of the interaction between stiffness and classic signaling pathways (regarding Comment 4) is intriguing. Incorporating downstream signaling and crosstalk would substantially strengthen the mechanistic understanding presented in the manuscript. While I understand the authors' intention to reserve some data for more in-depth analysis in future work, it is crucial to include some mechanistic analysis in the current manuscript to enhance its impact and completeness.

We agree with the reviewer that the manuscript would be more impactful when interactions between collagen crosslinking and biochemical signalling pathways are explored. As above, the significant additional figures and narrative that is required to present these data here would detract from the physical mechanism we would like to demonstrate. We regret we cannot add molecular mechanism to our physical mechanism of morphogenesis but hope that these data still confer significant new understanding of morphogenesis. We thank the reviewer for their enthusiasm regarding our sequencing-derived insights shared in our previous response.

Reviewer #3 (Remarks to the Author):

The editor has invited us to comment specifically on the mathematical modeling in the manuscript, welcoming comments on other aspects of the manuscript. Overall, the authors have derived, analyzed, and implemented a useful and appropriate model for analyzing self-propagating wave of bone morphogenesis. Firstly, they show how the equations can be derived from cell dynamics and mechanics. Then they use knowledge on the FKPP wave to derive conditions on the parameters that would produce a wave. The parameters are obtained from the experimental data and literature.

We thank the reviewers for the succinct summary and feedback. This input has significantly improved our manuscript.

Detailed comments for sections:

Theoretical model: The derivation looks all right, but can be written more clearly. Please be clear when referring other parts of the text, (e.g. “as described below” on line 630), and we found some typos that need correction (like velocities v_A and v_B).

We have added an equation reference for $k_i(\rho_i)$ to the sentence with “as described below” (Line 650). We have also rewritten the velocities v_A and v_B in vector form (Line 669).

The mixing of variable names A and B on the one hand and mnemonics derived from the cell names “osteoblasts” and “mesenchymal cells” (e.g., E_M and E_O) made the model difficult to read at some places. It would be helpful to use similar mnemonic symbols for A and B, e.g., c_O and c_M .

We thank the reviewers for pointing out this inconsistency and have renamed variables. In order to emphasize the general character of the model, we have renamed E_M and E_O to E_A and E_B .

Please explain the rationale for using Fick's law of diffusion with respect to the cells B.

We thank the reviewer for highlighting a need for clarification in this part of our methods and address this question with the following text now included:

Line 686-693: This approach is based on the assumption that in the absence of active processes (proliferation, differentiation), in the long-time limit the system becomes well-mixed, i.e. spatial inhomogeneities in cell numbers of A and B cells vanish and ϕ becomes spatially uniform. Provided there is no cell sorting (e.g. due to differential adhesion), random motion of cells would be sufficient to lead to mixing of different cell types. Hence we assume that in the absence of other processes, local differences in cell type composition will eventually relax to a homogeneous state where the local composition is uniform across space.

We felt that most attention should go to the section “Wave Solutions”.

- Could you explain why “osteoblast expansion dynamics follows that of a linear unstable wave”, l. 679? Even if you came to this conclusion based on observations in

the experiments, you cannot a rigid mathematical claim like this. An unstable wave is, as the name says, not stable and therefore not observable. The source that is cited, (67), discussed wave propagation towards an unstable homogeneous steady state, which does not imply that the wave itself is unstable. These comments also refer to the first few sentences in section "Wave Solutions": Not every system that shows a wave is FKPP.

We thank the reviewer for pointing out this issue in nomenclature and phrasing and apologize for the lack of explanation for the stated claim, which has been lost during a previous revision.

Rather than making a rigorous mathematical claim about wave properties, here our purpose is to rationalize the use of the FKPP framework for modelling osteoblast expansion. Our main arguments for using a FKPP wave to model the system follow two experimental observations:

- (1) Direct observation of differentiation events ahead of the osteogenic front in the form of isolated osteoblasts with only nuclear staining and no membrane staining (Fig. 2E-G). This is consistent with a picture that the mesenchymal progenitor state is unstable and fluctuations may induce cell differentiation into osteoblasts.
- (2) Measurement of the roughness coefficient of the osteogenic front, which agrees with that of the FKPP wave, as shown in Fig. S4D.

While these experimental findings do not rigorously preclude other mechanisms of expansion, they are consistent with and support the use of the FKPP framework for modelling osteoblast expansion in our system.

We have now significantly edited the supplementary text to include an explanation of the rationale behind the FKPP wave and removed terms relating to an unstable wave. As these changes are extensive we have not copied them here but they can be found in **lines 781-827**.

- Two mechanisms are described that are "capable of generating a FKPP wave". This wording indicates that only having "(1) a difference in proliferation rates" would be enough to create a FKPP wave, however it is necessary that $f(\phi=0)=0$, so k_D

needs to be dependent on ϕ . Consider starting with the conditions that are necessary for a FKPP wave (l. 688-689) and follow up with a mathematically rigorous approach that derives possible mechanisms that can generate such a wave. We think it would be wise to approach it the following way:

FKPP wave iff $f(1)=f(0)=0$, $f'(0)>0$ and $f'(1)<0$

$f(1)=0$ is already correct

So, we need

$f(0)=k_D(E(\phi=0)) = 0$

$f'(\phi)=(k_B-k_A)(1-\phi)-(k_B-k_A)\phi+(1-\phi) k_D'-k_D$

so

$f'(0)=(k_B-k_A)+k_D'(0)$

$f'(1)=-(k_B-k_A)-k_D'(1)$

We thank the reviewer for thoroughly checking our derivation and suggesting an alternative presentation of the results. We have now rederived the results according to the reviewer's suggestion, which coincidentally led to another condition on k_D that was previously left out, namely $k_D[\phi=1] > 0$. The form for k_D previously chosen in the simulations is consistent with this additional condition, so no additional simulations were required.

- Lastly, it may be better to move section "Wave Solutions" to after section ("Full System"), because you need the derived equations and information like boundary conditions to do your FKPP analysis.

We appreciate this suggestion. We have revised the order of the paragraphs such that "Full System" now immediately follows the model derivation, and "Mechanical Feedback" and "Wave Solutions" are placed after this.

Comments on other aspects of the manuscript:

- At the end of the first introductory paragraph, the authors write: "Indeed, the suffix - taxis which means directional movement, is entirely synonymous with cellular migration in cell and developmental biology." We do not agree with this statement. Taxis specifically means directional migration along environmental (or self-generated) gradient, as the authors state in the previous sentences. However, cellular migration also includes *random* cell migration (e.g. of neural crest cells or immune cells) and directed cell migration *not* along a specific gradient (sometimes also seen in collective

neural crest cell migration, or e.g. in crescent shaped keratocytes). This remark also applies to l. 39.

We thank the reviewers for their suggestion and have removed these statements from the introduction.

- Please integrate the mathematical model better within the main text. The part on modeling starts with (l. 149) “We simulated the model with realistic biological parameters”, followed by the model predictions. Please clarify on what biological assumptions the model is based, and how these related to the experimental findings. We recommend that the process descriptions (l. 629-633) as well as the mechanical feedback mechanism (l. 644 and onwards) given in the supplementary material are moved to the main text. Then later on in the main text the paragraph starting with ‘Therefore, we proposed that the osteogenic wave of our model (Fig. S4C-F; Supplementary Text) ...’ could be made more specific by highlighting specific predictions on the role of mechanical feedback and discussing how exactly these agree (or not) with the data.

We thank the reviewers for their suggestions as they have improved the clarity of our rationale and the link from our modelling to experiments. We have clarified our model assumptions and details as indicated by the reviewer to the results as follows:

Line 147-155: Our model includes two processes that modulate local cell concentrations: (1) proliferation and cell death give rise to effective reproduction rates that depend on local cell densities, (2) cell differentiation, whereby progenitors are converted to osteoblasts at a rate which can depend on other (dynamic) variables of the system. Furthermore, we modelled forces capable of generating cellular flows and arising from the balance between pressure gradients, viscous forces, and friction. To model the observed inhomogeneities in stiffness, we let the tissue stiffness depend locally on cell type, to reflect differential rates of collagen production whereby differentiated osteoblasts generate a stiffer environment than undifferentiated mesenchyme.

Line 159-165: Our model generates wave solutions which recapitulate the expansion of the osteoblast domain, with differential velocities for the osteogenic front and for individually tracked cells. To obtain all model parameters, we estimated the stiffnesses and homeostatic cell densities of mesenchyme and osteoblast tissue as well as the

*diffusion constant directly from experimental data and estimated the viscosity, friction coefficient and relaxation time of the net division rate from literature (Table S1). By fitting a single parameter (describing the relation between differentiation and stiffness) in numerical simulations, we simultaneously fit the experimentally measured values obtained from our live imaging analyses and *Osx1-GFP::Cre* intensity values extracted from fixed labeled stage series skull caps (Fig. 3B; Fig. S5).*

We have added a more specific statement, as suggested, to clearly specify that our model predicts that the mechanical forces generated by a collagen or stiffness gradient is the driving force of morphogenesis as follows:

Line 168-170: Therefore, we proposed that the osteogenic wave is instead driven by the aforementioned mechanical feedback, whereby a self-generated collagen gradient generates both pushing forces from emergent pressure gradients, as well as an osteoblast differentiation wave arising from a stiffness-dependent differentiation rate.

- In section “Perturbing the stiffness gradient changes bone size” we would prefer that the model predictions are shown first (after l. 175) after which it can be appreciated to what extent the model predictions match the data. Currently, Fig. 4O-P do not give sufficient specific information as to whether and how the model predictions match the data (i.e., where can we see in this figure that ‘Our model predicts that a greater difference in tissue stiffness between the bone center and front would promote the expansion of frontal bones.’?). Possibly this could be solved by a more detailed description in the caption and reference to figure 4P after the statement ‘our model predicts that...’.

We thank the reviewer for highlighting the need for further clarification of our models prediction in relationship to our experimental validations. We have added to this section to address this deficit as follows in highlighted text:

Line 187-207: Our model predicted that a stiffness gradient is sufficient to drive both cell motion and differentiation toward the midline. The velocity of motion and differentiation would then be dependent on the slope of the gradient. The greater the stiffness gradient, the faster cells would move and differentiate (See Supplementary Text). To test this prediction in vivo, we performed two perturbations. In the first, we performed live imaging on E13.75 skull caps in which the stiffer bone center was

excised (Fig. 4A). Our model lead us to predict that bone expansion should be halted in the absence of the stiff bone center. We found little bone expansion throughout our imaging experiments suggesting that the stiffer bone center contributes to morphogenesis of the frontal bone (Fig. 4B). In our second perturbation, we aimed to increase the stiffness gradient which would increase pushing forces from the bone center to increase the rate of bone expansion. We chemically blocked collagen crosslinking by feeding pregnant dams with Beta-Aminopropionitrile (BAPN), an irreversible competitive inhibitor of LOX (38-43). Loss of crosslinking would allow for collagen fiber deformation upon physical interactions with cells or longer-range forces such as stretch from the underlying expanding brain. Indeed, we found that fiber area was perturbed in the newly differentiating bone (Fig. 4F, G) which led to poorer mineralization at the end of skull morphogenesis as expected (Fig. S6). As we predicted, BAPN-treatment increased the stiffness gradient by reducing tissue stiffness at the osteogenic front (Fig. 4D-I) although stiffness in the bone center mildly increased (Fig. 4I). Consistent with our prediction, we found significantly larger frontal bones toward the end of skull expansion (Fig. 4J-L). To test whether increased differentiation contributed to these larger bones we measured the intensity of the *Osx1-GFP::Cre* reporter as a proxy for osteoblast "age" as before.

In the supplementary text, we have further clarified how the stiffness gradient affects the speed of the wave:

Line 852-857: Secondly, both the advection velocity as well as the FKPP wave front velocity increases with larger differences in stiffness across the system (i.e. upon increasing $E_B - E_A$). In the case of the advection velocity v_A , this is due to the mechanical effect of increased pressure gradient across the tissue, since the pressure is proportional to stiffness (Eq. 13). This in turn leads to an increase in the pressure-derived force in Eq. 20, which in the inviscid limit is directly proportional to the advection velocity. In the case of the FKPP velocity v_{FKPP} , this arises from an increase of the differentiation rate with larger stiffness differences as described in Eq. 30.

Reviewer #4 (Remarks to the Author):

I co-reviewed this manuscript with one of the reviewers who provided the listed reports. This is part of the Nature Communications initiative to facilitate training in peer

review and to provide appropriate recognition for Early Career Researchers who co-review manuscripts.

We are pleased that reviewer 4 was able to participate in reviewing our manuscript and we are grateful for their critical thinking and valuable suggestions. We hope that we have sufficiently addressed your comments in our revised manuscript.

REVIEWER COMMENTS Revision 3 NCOMMS-24-14942B

Reviewer #3 (Remarks to the Author):

The authors have significantly clarified the integration of the mathematical model in the main text, and they have improved the supplementary text. In this new version, the story has become stronger from a mathematical viewpoint. There are still a few small corrections to be made:

II. 838-840:

Add: "Since $E_B > E_A$ "

"Above constraints": please clarify to what constraints this refers to.

We thank the reviewer for their suggestion and have changed "above constraints" into "constraints for a FPKK wave, Eqs. 23 and 26".

Add some information when you introduce new variables or parameters. What is α and what conditions does it have to comply with? (e.g. $\alpha > 0$)

We thank the reviewer for this suggestion and have added information about new parameters in the following lines:

Line 842-843: "Here, α represents a proportionality constant quantifying how sensitive the differentiation rate is to change in stiffness. Since we assume that stiffness E_{ϕ} cannot be lower than the mesenchymal stiffness E_A , for any $\alpha > 0$ we obtain a positive differentiation rate $k_D > 0$. Together with Eq. 21, this choice of k_D (Eq. 27) implies that the total differentiation rate takes the form corresponding exactly to the classical FKPP equation"

(825- 829) add "," and "." after equations

(864) Change " E_O " and " E_M " to the new notation E_A and E_B and check the rest of the text for remaining instances.

We thank the reviewer for pointing out these typos and have corrected them as suggested.

Reviewer #3 (Remarks on code availability):

There is a data availability statement, but it is disturbingly inadequate. The authors need to have a look at it. They write that the code is available "through the Zenoba (sic) code repository". (Googling 'Zenoba' bring me to a brand of sunscreen). The link brings us to the bioRxiv-preprint of this manuscript, where we did not find any code.

After some googling, we found a Zenodo repository at <https://zenodo.org/records/8246065>. This repository contains data and Mathematica code, but no instructions. We tried to run two of the Mathematica-files without success. 'Tissue_expansion_model_4_FD_solver_loop.nb' starts by attempting to run some data from one of the authors home folders. We briefly looked into changing the 'LoadFolder', but it is quite unclear to what folder we need to point LoadFolder, because the in the Zenodo 'Data' folder do not align with those mentioned in the Mathematica File. There is no 'simulation_set1/finished_sims_A/data' and in the existing folders we find files mathcing '*_phi_*', but none matching '*_V_*' or '*_rho_*'.

We also checked 'Tissue_expansion_simplified_model_2022_12.nb'. The code was hard to read or to align with the equations in the manuscript. We attempted evaluating all expressions up to 'ndsol = NDSolve' (which is expected to numerically solve the actual model), but this expression failed with "NDSolve::ndcf: Repeated convergence test failure at t == 0.0004`; unable to continue."

We would kindly ask the authors to provide a 'README' file, including some instructions for running the model and reproducing the modeling results presented in the manuscript (i.e. indicating specifically for the modeling figures how they can be reproduced).

We apologize for the lack of information on code and data availability of our work. While we have indeed included all relevant code for this manuscript, the Zenobo repository was not intended for review without further instructions and therefore we fully understand your inability to get our code to work.

We have now reorganized our code and included ample instructions for how to run the relevant code we used for producing the graphs in the manuscript. In particular, we have prepared a cleaned up version of the Mathematica code used to simulate the tissue mathematical model, along with detailed instructions and notes.

As our code runs on proprietary software, we were unable to use Code Ocean for easy access and reproduction. Instead, we have uploaded the improved code and instructions to a new Zenodo repository (since Zenodo does not allow for editing of already published repositories): <https://zenodo.org/records/12960584>.

Within the repository, you will find a readme.txt file explaining the main structure of the repository. The “Code” subfolder is a copy of the GitHub repository at <https://github.com/YitengDang/SkullWave> and contains all code developed and used during this project. The “Data” subfolder contains data used to generate the main plots of the paper. The README.md file of the GitHub repository contains further description of the various subfolders and files found in the repository. Finally, detailed information on running the simulations has been added to the Mathematica script at “./Code/For_publication/Tissue_model_simulator.nb”. We have added this information to our data availability section (line 746)

We hope that with the improved code and instructions, the reviewers will be able to run our code and reproduce our results.

Reviewer #4 (Remarks to the Author):

We thank reviewer #4 for their continued contribution in reviewing the paper.